# Exome Sequencing Revealed a Novel Splice Site Variant in the *CRB2* Gene Underlying Nephrotic Syndrome

**DOI:** 10.3390/medicina58121784

**Published:** 2022-12-04

**Authors:** Anam Simaab, Jai Krishin, Sultan Rashid Alaradi, Nighat Haider, Muqadar Shah, Asmat Ullah, Abdullah Abdullah, Wasim Ahmad, Torben Hansen, Sulman Basit

**Affiliations:** 1Department of Biochemistry, Faculty of Biological Sciences, Quaid-I-Azam University (QAU), Islamabad 45320, Pakistan; 2Department of Pediatrics, Pakistan Institute of Medical Sciences, Shaheed Zulfiqar Ali Bhutto Medical University, Islamabad 44000, Pakistan; 3Department of Laboratory Sciences, Al Wajh General Hospital, Al-Wajh City 48722, Saudi Arabia; 4Novo Nordisk Foundation Center for Basic Metabolic Research, Section of Metabolic Genetics, Faculty of Health and Medical Sciences, University of Copenhagen, 2200 Copenhagen, Denmark; 5Center for Genetics and Inherited Diseases, Taibah University, Medina 42318, Saudi Arabia

**Keywords:** nephrotic syndrome, whole exome sequencing, novel splice site variant, *CRB2*

## Abstract

*Background and Objectives*: Nephrotic syndrome (NS) is a kidney disease where the patient has a classic triad of signs and symptoms including hypercholesterolemia, hypoalbuminemia, proteinuria (>3.5 g/24 h), and peripheral edema. In case of NS, the damaged nephrons (structural and functional unit of the kidney) filter unwanted blood contents to make urine. Thus, the urine contains unwanted proteins (proteinuria) and blood cells (hematuria), while the bloodstream lacks enough protein albumin (hypoalbuminemia). Nephrotic syndrome is divided into two types, primary NS, and secondary NS. Primary NS, also known as primary glomerulonephrosis, is the result of a glomerular disease that is limited to the kidney, while secondary NS is a condition that affects the kidney and other parts of the body. The main causes of primary NS are minimal change disease, membranous glomerulonephritis, and focal segmental glomerulosclerosis. In the present study we recruited a family segregating primary NS with the aim to identify the underlying genetic etiology. Such type of study is important in children because it allows counseling of other family members who may be at risk of developing NS, predicts risk of recurrent disease phenotypes after kidney transplant, and predicts response to immunosuppressive therapy. *Materials and Methods*: All affected individuals were clinically evaluated. Clinical examination, results of laboratory tests, and biopsy investigations led us to the diagnosis. The next-generation sequencing technique (whole-exome sequencing) followed by Sanger sequencing identified a novel homozygous splice site variant (NM_173689.7: c.941-3C>T) in the CRB2 gene. The variant was present in a homozygous state in the affected individuals, while in a heterozygous state in phenotypically normal parents. *Results*: The study expanded the spectrum of the mutations in the gene CRB2 responsible for causing NS. *Conclusions*: In addition, the study will also help in genetic counseling, carrier testing, and prenatal and/or postnatal early diagnosis of the disease in the affected family.

## 1. Introduction

Podocytes are highly polarized and specialized epithelial cells critical for renal glomerular filtration. These cells perform their function via interdigitated foot processes that are connected by a slit diaphragm. Any disruption in this foot process organization leads to abnormal renal filtration resulting in symptoms of nephrotic syndrome [1,2,3,4]. 

Nephrotic syndrome is the most common glomerular defect of childhood characterized by nephrotic-range proteinuria which may be defined as ≥40 mg/m^2^/h or urine protein/creatinine ratio ≥200 mg/mL or urine dipstick showing 3+ protein, serum albumin < 25 g/L, and edema [5]. In a cross-sectional study, Najam-ud-Din et al. [6] found that oedema was the most common presenting complaint, while oliguria, abdominal tenderness, fever, hematuria, uraemia and thrombosis were found in descending orders in a cohort of 360 Pakistani patients with suspected nephrotic syndrome [6]. Children with nephrotic syndrome can develop life-threatening complications, including infection and thrombosis, hypovolemia, and acute kidney injury (AKI). AKI is common in children hospitalized with nephrotic syndrome and should be deemed the third major complication of nephrotic syndrome in children, in addition to infection and venous thromboembolism. Risk factors for AKI include steroid-resistant nephrotic syndrome, infection, and nephrotoxic medication exposure. Children with AKI have longer hospital lengths of stay and increased need for intensive care unit admission [7]. The disease became life-threatening when manifested as NS in the first year of life (NSFL). Response to treatment is observed in ≤80% of cases in older children, that is why standard treatment protocols are also used widely for older children. Clinical decision-making in children with NSFL still remains challenging. Descriptively, NSFL has been classified as congenital NS (CNS), manifesting in utero or during the first 3 months of life, infantile NS (INS), with onset between 4 months and 1 year of age, and childhood NS in children older than 1 year [8].

The United States reported the annual incidence rate of the nephrotic syndrome as two to seven cases per 100,000 children younger than 16 years, while the cumulative prevalence rate is about 16 cases per 100,000 individuals [8,9]. 

Nephrotic syndrome is a high heterogeneous kidney disorder showing autosomal as well as X-linked inheritance [10]. Until now, 66 genomic loci have been associated with human nephrotic syndrome. Pathogenic variants in 58 genes have been previously reported to cause SRNS [11,12]. Recently, Basit et al. [10] reported a family of Saudi origin segregating X-linked SSNS caused by a pathogenic variant in CENPI. Rest of the SSNS-associated loci yet need to be investigated for disease-causing genes. Due to huge genetic heterogeneity and extreme rarity of the disorder, whole exome sequencing is required to identify the genetic cause of the phenotypes.

Crumbs homolog 2 (CRB2) is one of the three mammalian homologs of the Drosophila Crumbs (Crb) protein which regulates the differentiation and polarization of epithelial cells [13]. CRB2 protein is encoded by CRB2 gene spanning around 22.49 kb region on chromosome 9q33.4. The longest transcript (NM_173689.7) of CRB2 contains 13 coding exons translating into 1285 amino acid protein (Figure 1b). CRB2 is mainly expressed in the retina, brain, and kidney [14]. Previously, homozygous and/or compound heterozygous sequence variants in the gene have been associated with NS [14,15,16]. Ebarasi et al., (2015) reported four families of Turkish and European union affected with NS, and identified five recessive pathogenic variants (p.Cys620Ser, p.Arg628Cys, p.Gly1036Alafs*43, p.Cys629Ser, p.Arg1249Gln) in *CRB2* underlying the disease phenotypes. The authors reported that zebrafish *crb2b* is required for podocyte foot process arborization, slit diaphragm formation, and proper nephrin trafficking. Since CRB2 is required for epithelial apical-basal differentiation, the loss of CRB2 function causes podocyte alterations leading to kidney diseases.

Here, we presented the clinical and genetic investigation of a Pakistani family segregating NS in an autosomal recessive manner. Whole exome sequencing (WES) followed by Sanger sequencing revealed a novel homozygous sequence variant in the gene CRB2 segregating with the disease phenotypes in the family.

## 2. Materials and Methods

### 2.1. Ethical Approval

The Institutional Review Boards (IRB) of Quaid-i-Azam University Islamabad and Shaheed Zulfiqar Ali Bhutto Medical University Islamabad, Pakistan have given permission to conduct this study. Furthermore, the participating individuals and/or their legal guardians signed informed consent forms to participate in the proposed research study.

### 2.2. Genomic DNA Extraction and Biochemical Analysis

Peripheral blood samples from available healthy and affected individuals of the family were collected in ethylene diamine tetra acetic acid-containing tubes. Genomic DNAs from the collected blood samples were extracted using QIAamp DNA Mini Kit (Qiagen, Hilden, Germany) and subsequently quantified by Nanodrop-1000 spectrophotometer (Titertek Berthold, Thermo Fisher Scientific 3411 Silverside Road Bancroft Building, Suite 100 Wilmington, DE 19810 USA). 

Partial thromboplastin time (PTT) and activated partial thromboplastin time (aPTT) were performed to characterize coagulation of the blood. Protein C, S and antithrombin III were also tested and found to be normal. A biopsy was performed under general anesthesia under strict aseptic measures in operation theatre. An automated Tru-Cut biopsy needle was used to take biopsy under ultrasound guidance. Two samples were taken, one for immunofluorescence and one for histopathology.

### 2.3. Whole Exome Sequencing and Variant Selection

To identify a potential candidate gene, genomic DNA of one of the affected individuals (V-4) was used to perform whole exome sequencing. The sample was prepared using Agilent SureSelect Target Enrichment Kit by following the manufacturer’s guide. The libraries were sequenced with Illumina HiSeq 2000/2500 sequencer. The BWA Enrichment application of BaseSpace (Illumina Inc. 5200 Illumina Way|San Diego, CA, 92122, USA) was used to analyze the reads. Alignment was performed with Burrows-Wheeler Aligner (BWA) and the variants were called with Genome Analysis Toolkit (GATK). All the called variants were annotated with Illumina Variant Studio v2.2. Based upon the inheritance pattern observed in the pedigree of the present family, we filtered the biallelic variants. Exome sequence data was carefully analyzed and the presence of all suspected homozygous variants was checked in the public databases [dbSNP (https://www.ncbi.nlm.nih.gov/snp/ accessed on 10 February 2022), 1000 Genomes browser (https://www.ncbi.nlm.nih.gov/variation/tools/1000genomes/ accessed on 10 February 2022), Exome Variant Server (https://evs.gs.washington.edu/EVS/ accessed on 10 February 2022), gnomAD (https://gnomad.broadinstitute.org/ accessed on 10 February 2022)], and the exome data of 53 healthy controls collected from different ethnic groups of Pakistan.

### 2.4. Primer Designing and Sanger Sequencing

Based upon previously reported studies, demonstrating the association of CRB2 with NS, the splice site variant in the gene identified in exome data of the patient (V-4) was considered as the strongest candidate for segregation analysis. Thus, the variant was Sanger-sequenced in the DNA of affected and unaffected members of the family. Primers were designed for specific exon-intron border using online Primer 3 software (https://bioinfo.ut.ee/primer3-0.4.0/ accessed on 10 February 2022). Basic Local Alignment Search Tool (http://www.ensembl.org/Multi/Tools/Blast accessed on 10 February 2022) was used to examine primers specificity. In silico PCR for designed primers was performed using an InSilico PCR Tool (https://www.genome.ucsc.edu/cgi-bin/hgPcr accessed on 10 February 2022). Primers were amplified using DNA of affected and unaffected individuals following standard PCR conditions. Amplified products were run on 2 % agarose gel with a 100 bp DNA ladder. Specific PCR products were purified using a commercially available kit (Axygen^®^ AxyPrep PCR Clean-Up kit, Union City, CA, USA). Amplified products were then sequenced by Sanger sequencing using BigDye Terminator v3.1 Cycle Sequencing Kit (Life Technologies, Carlsbad, CA, USA) according to manufacturer’s instructions. The Sanger sequencing data were searched for disease-causing variants by comparing them with reference gene sequence, downloaded from Ensembl Genome Browser (http://www.ensembl.org/index.html accessed on 10 February 2022). Pathogenicity of the variation was predicted using online variant effect prediction tools, including MutationTaster (http://www.mutationtaster.org/ accessed on 26 June 2022), VarSome (https://varsome.com/ accessed on 26 June 2022), and splice site prediction software, varSEAK (https://varseak.bio/ accessed on 26 June 2022). Analysis of the flanking sequences of the variant for predicting alternative splice site was performed using Alternative Splice Site Predictor (ASSP) (http://wangcomputing.com/assp/ accessed on 26 June 2022).

## 3. Results

### 3.1. Clinical Features 

The affected family was recruited from the nephrology ward of the children’s hospital at Pakistan Institute of Medical Sciences (PIMS) Islamabad, Pakistan. On the basis of clinical features, biopsy examination, and response to treatment, the affected individuals of the family were diagnosed to segregate nephrotic syndrome. The family has seven affected individuals (IV-5, IV-8, IV-9, IV-10, V-4, V-7, V-8) in three loops in two different generations (Figure 1). Parents of the affected individuals are first cousins in all three loops. Three affected siblings (V-4, V-7, V-8) and their parents (IV-3, IV-4) participated in the present research project.

The first affected sib (V-4) was female and was the fourth product. She developed generalized body swelling at the age of 5 years and was treated with deltacortil (a steroid used to treat multiple medical conditions including NS). Renal biopsy revealed focal segmental glomerulosclerosis. Thus, the child was diagnosed with congenital nephrotic syndrome. There was a partial response to deltacortil so labeled as steroid-resistant nephrotic syndrome.

The second affected sib (V-7; Figure 1b) was the seventh product who developed generalized swelling at the age of 1 year. Laboratory tests performed on patient’s urine revealed urine protein/creatinine ratio > 200 mg/mL which is in the range of NS. He was labeled steroid-resistant nephrotic syndrome after multiple admissions and poor response to deltacortil. 

The third patient (V-8) was the eighth product among the sibship of nine. She developed generalized body swelling at the age of 2 years and was treated with deltacortil. The patient was diagnosed with NS based on the results of laboratory tests, generalized edema, and previous family history of the disease. She was treated with deltacortil (Table 1). There was a response to deltacortil within 2 weeks but she again developed edema while tapering the deltacortil so labeled as steroid-dependent nephrotic syndrome and was given cyclosporin along with ACE (Angiotensin-converting enzyme) inhibitors.

Biopsy was advised to the other two siblings (V-7, V-8) but declined by their parents. All the children were born at home through spontaneous vertex delivery with normal antenatal history. They were vaccinated according to the Expanded Programme on Immunization (EPI) and were developmentally normal (Table 1).

### 3.2. Genetic Description

In the current investigation, the family showing autosomal recessive inheritance of steroid-resistant nephrotic syndrome belongs to the Pakhtunkhwa province of the country.

Whole exome sequencing of one of the affected individuals (V-4) revealed 113,640 single nucleotide variations/small indels among which 15,146 were nonsynonymous/splice sites. There were 5537 homozygous and 9609 heterozygous nonsynonymous/splice site sequence variants in the exome file of the affected individual. Based upon the inheritance pattern of the disease and consanguinity among parents of the affected individuals, only homozygous sequence variants were prioritized for further analysis. All the sequence variants previously reported in any normal human genome variation database including dbSNP, 1000 Genomes browser, Exome Variant Server (EVS), and gnomAD with minor allele frequency more than 0.01 were excluded from the exome file. The final list of variants after the prioritization process includes five homozygous variants. Genes were prioritized based on their expression and function in the tissue, and previous association with the disease in the literature. An acceptor splice site variant (c.941-3C>T) in the gene CRB2 was identified in the exome of the affected individual. Based upon CRB2 function, expression and previous association with kidney disorder, a variant in the gene was hypothesized to be the strongest candidate variant. Sanger sequencing found the variant in homozygous state in all the three affected individuals while heterozygous in parents (Figure 2). The identified variant was not found in 100 ethnically-matched control individuals. The identified variant was predicted to be disease-causing by MutationTester. VarSome software predicted the identified variant with uncertain significance. Interestingly, splice site prediction software; varSEAK classified the identified variant as “CLASS 4” having a likely splicing effect (Figure 3).

## 4. Discussion

The study presented here describes the clinical and genetic characterization of a consanguineous family segregating nephrotic syndrome in an autosomal recessive manner. Most of the NS-related features observed in affected members in our family were similar to those reported previously in other cases. In previous cases, most of the patients have facial edema followed by generalized body edema, particularly in the feet and ankles ([3], http://www.hgmd.cf.ac.uk/ac/all.php accessed on 26 June 2022). In contrast, the present patients, showed generalized body edema at the onset of the disease. Age at the time of onset of disease varies among present affected individuals. 

Previous studies show that more than 80% of children with childhood NS have Steroid-sensitive nephrotic syndrome (SSNS). On the other hand, approximately 20% of children with childhood NS and most children with congenital or infantile NS have steroid resistant nephrotic syndrome (SRNS) [8]. In the present study, two of the affected individuals (V-4, V-7) have SRNS, while the third patient (V-8) presents phenotypes of SSNS. In NS, it has been observed that some patients are steroid dependent initially, and then become steroid resistant latter on. The present patient (V-8) needs a further follow up to check whether she stays with the same condition, or it changes to steroid-resistant. Some variability in the severity of phenotypes of the present patients was also observed. Variability in the severity of phenotypes and age onset of the disease might be due to variable expressivity in patients due to involvement of SNPs in modifier genes, epigenetic factors or measures that can slow down the progression of kidney disease such as control of hypertension.

The CRB2 gene encodes a 1285 amino acids polypeptide, which is composed of three extracellular laminin G-like domains and 15 extracellular EGF-like domains. Crumbs homolog (CRB) proteins are the main polarity determinants of apical and basolateral membrane domains in the epithelial cells. In kidneys, it regulates differentiation and epithelialization of podocyte cells which are the main regulators for renal glomerulus filtration [1]. Until now, total thirty-two variations have been identified in the gene causing different types of kidney disorder [17,18,19] http://www.hgmd.cf.ac.uk/ac/all.php accessed on 26 June 2022). Here, we have identified the very first familial case of Pakistani origin of nephrotic syndrome caused by a novel homozygous splice site variant in the gene.

The novel homozygous sequence variant (c.941-3C>T) identified in the DNA of the affected individuals in the present family locates at the splice acceptor site of intron 5 of CRB2. The identified variant is likely to disrupt normal splicing of its mRNA and hence protein functions. The mis-splicing is predicted to cause retention of intron 5 in the CRB2 mature mRNA thus, leading to frameshift of the protein. Analysis of the flanking sequences of the variant (c.941-3C>T) suggests the possible use of cryptic splice acceptor site, located 189 bp downstream within exon 6, which could lead to the production of a mutant protein with residual function. The identified variant is predicted to lead to abnormal or absence of normal CRB2 protein function, due to which glomeruli will have abnormal podocytes and filtration leading to the disease phenotypes.

In conclusion, we have demonstrated characterization of an extended consanguineous family of Pakistani origin segregating NS in an autosomal recessive manner. Using Whole exome and/or Sanger sequencing technology, sequence analysis of the genome of the affected and unaffected individuals revealed a novel homozygous disease-causing variant c.941-3C>T in the CRB2 gene. The present findings not only expanded mutational spectrum in the CRB2 gene but will also be helpful in carrier testing, genetic counseling, and facilitating diagnosis of NS cases in the Pakistani population. The study may bring awareness regarding the disease in the afflicted families to enable them to manage the disease in a better way, hence increasing the life expectancy of patients.

## Figures and Tables

**Figure 1 medicina-58-01784-f001:**
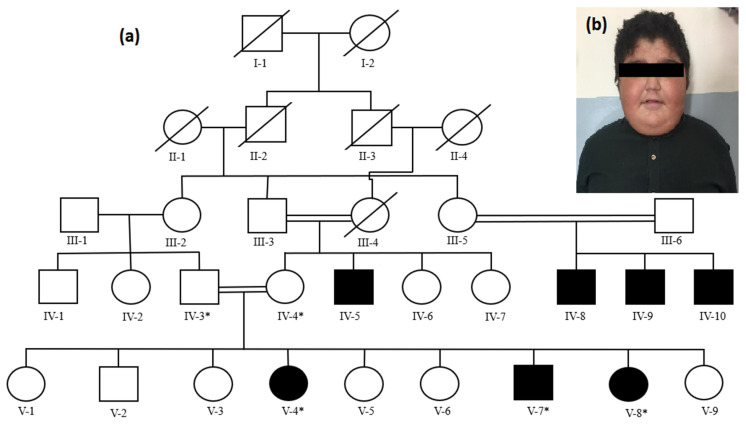
Pedigree of a Pakistani family showing inheritance of nephrotic syndrome in an autosomal recessive form (**a**). Asterisk (*) sign above squares and circles shows availability of the individuals for participating in the study. Clinical presentation of an affected individual (V-7) showing generalized body edema (photograph of the affected individual is shared with the consent of parents) (**b**).

**Figure 2 medicina-58-01784-f002:**
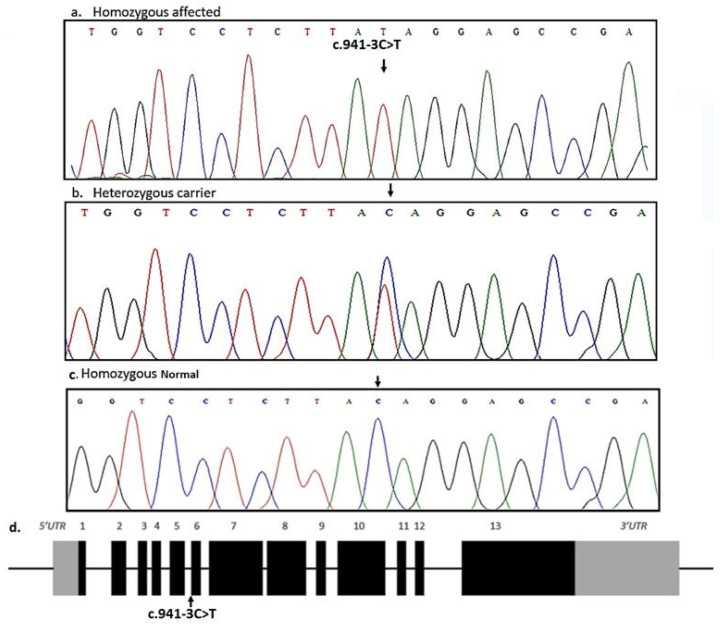
Sequencing results of intron5_exon6 border in CRB2 (5′–3′). A homozygous mutation (c.941-3C>T) in the affected individuals is indicated by an arrow (**a**). c.941-3C>T in CRB2 in heterozygous state in parents of the affected individuals (**b**). Wild type sequence of intron5_exon6 border in CRB2 in healthy control individuals (**c**). Structure of the CRB2 gene showing coding exons and non-coding introns, 3′ and 5′ UTRs, and position of the identified mutation in splice acceptor site of intron 5 (**d**).

**Figure 3 medicina-58-01784-f003:**
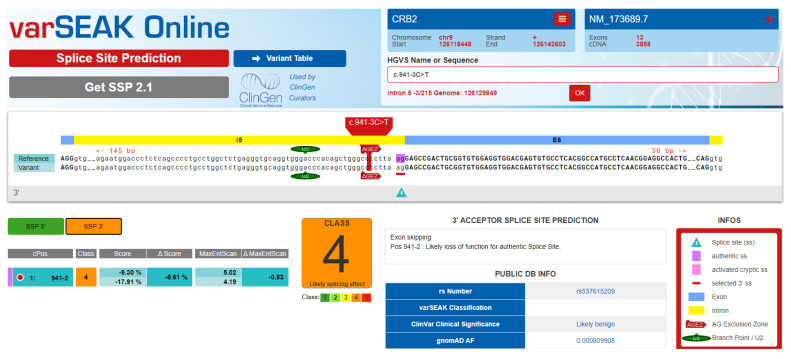
Prediction of VarSEAK software.

**Table 1 medicina-58-01784-t001:** Laboratory findings of affected individuals.

Parameters	Patient V-4	Patient V-7	Patient V-8	Reference Range
Age	13	11	8	
Weight	63 kg	36 kg	29 kg	
Height	154 cm	139 cm	124 cm	
**Blood routine tests**				
Erythrocytes, m/mm^3^	5.3	4.9	5.7	4.7–6.1
Leucocyte, 10^9^/L	6.73	7.32	5.92	4.0–10.5
Platelet count, 10^9^/L	345	298	166	150–400
Hemoglobin, g/L	13.2	11	10.4	12.5–16.1
Neutrophil, (%)	63	73	59	54–62
Lymphocyte (%)	33	27	30	25–33
**Urine routine tests**				
Specific gravity	1.010	1.006	1.009	1.005–1.025
Urine protein	300 mg/dl	700 mg/dl	650 mg/dl	<150 mg/dl
Urinary occult blood	3	0	2	<4 RBC/HPF
Urine glucose	<130	<130	<130	<130 mg/dl
Urine ketone bodies	Nil	nil	Nil	nil
Urine creatinine, mmol/L	7	8.2	6.9	6–13
24-h UPE, mg/d	<100	<100	<100	<100 mg/m^2^/day
Urinary β2- microglobulin, mg/L	0.1	0.2	0.13	<0.3
**Immunology**				
Immunoglobulin A, g/L	104	188	204	33–236
Immunoglobulin G, g/L	788	1023	987	608–1572
Immunoglobulin M, g/L	69	123	215	52–242
Complement C3, g/L	98	102	173	88–201
Complement C4, g/L	32	39	22	15–45
**Serum chemistry**				
Total protein, g/L	6.8	7.2	6.3	6.0–8.3
Albumin, g/L	2.5	2.3	2.4	3.5–5.6
Globulin, g/L	2.5	2.9	3.3	2.0–3.5
Serum creatinine, mmol/L	0.36	0.64	0.73	0.31–0.88
**Blood coagulation tests**				
Prothrombin time	11.3	10.7	11.9	10.1–12
Activated partial thromboplastin time, seconds	33	35	29	26–36
Prothrombin time-international normalized ratio	0.81	0.91	0.89	0.8–1.1
Fibrinogenic, g/L	159	236	342	156–400
Renal biopsy findings	focal segmental glomerulosclerosis	-	-	

## Data Availability

Data available on request from the authors.

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
