# Peer review of "Exome Sequencing Revealed a Novel Splice Site Variant in the CRB2 Gene Underlying Nephrotic Syndrome"

_medicina, 2022, doi:10.3390/medicina58121784_

Round 1

Reviewer 1 Report

In this paper, the authors describe the occurrence of nephrotic syndrome (NS) in three siblings and their family members. They claim that a new variant in the CRB2 gene (c.941-3C>T) is responsible for NS in the patients.

The authors present the pedigree of the family going back 5 generations. NS has been reported in generations 4 and 5. In all instances, parents of NS patients were related. In addition, the authors report the results on genetic testing in three patients and their parents. A splice variant (c.941-3C>T) in the gene CRB2 was present in a homozygous state in the patients and in a heterozygous state in the parents. Overall, the results are consistent with the claim that the variant reported in CRB2 is the cause of NS. Addressing the following points may help strengthen the paper further.

1.     Firstly, the paper doesn’t seem to clearly state that CRB2 mutations have been previously associated with NS in patients and that the association between CRB2 variants and epithelial development was validated in a zebra fish model (Ebarasi et al). Ebarasi et al has been cited but the association between CRB2 variants and NS described in the paper can be more clearly made. This will support the claims made in this paper further

2.     In the genetic testing segment, the authors state that they discovered 5 homozygous variants. It would be helpful if the authors include the list of the genes are variants at least in the supplemental information. Were the parents tested for the other 4 variants?

3.     Figure 2e seems to be a screenshot. The font is too small to be legible. It would be beneficial to readers if the image is modified to be more readable.   

Author Response

Reviewer 1

Comments and Suggestions for Authors

In this paper, the authors describe the occurrence of nephrotic syndrome (NS) in three siblings and their family members. They claim that a new variant in the CRB2 gene (c.941-3C>T) is responsible for NS in the patients.

The authors present the pedigree of the family going back 5 generations. NS has been reported in generations 4 and 5. In all instances, parents of NS patients were related. In addition, the authors report the results on genetic testing in three patients and their parents. A splice variant (c.941-3C>T) in the gene CRB2 was present in a homozygous state in the patients and in a heterozygous state in the parents. Overall, the results are consistent with the claim that the variant reported in CRB2 is the cause of NS. Addressing the following points may help strengthen the paper further.

  1. Firstly, the paper doesn’t seem to clearly state that CRB2 mutations have been previously associated with NS in patients and that the association between CRB2 variants and epithelial development was validated in a zebra fish model (Ebarasi et al). Ebarasi et al has been cited but the association between CRB2 variants and NS described in the paper can be more clearly made. This will support the claims made in this paper further

Response: Thank you for the helpful comments. We have added a paragraph regarding involvement of pathogenic variants in CRB2 underlying Nephrotic Syndrome.

  1. In the genetic testing segment, the authors state that they discovered 5 homozygous variants. It would be helpful if the authors include the list of the genes are variants at least in the supplemental information. Were the parents tested for the other 4 variants?

Response: We have added the additional four variants in the supplementary table 1.

  1. Figure 2e seems to be a screenshot. The font is too small to be legible. It would be beneficial to readers if the image is modified to be more readable.   

Response: We have added a revised figure with better quality.

Reviewer 2 Report

The manuscript presented by the Authors entitled "Exome Sequencing Revealed a Novel Splice Site Variant in the 2 CRB2 Gene Underlying Nephrotic Syndrome" has potential but can be improved. The objectives are stated clearly. The results presented are relevant. The bibliography is up-to-date. The following are my comments describing these issues.

INTRODUCTION

The Authors should deepen these aspects in the introduction:

Point 1: Various complications have been reported in children with NS, including infections, thromboembolism, hypovolemia, and acute kidney injury (doi: 10.2215/CJN.06620615). AKI is common in children hospitalized with nephrotic syndrome and should be deemed the third major complication of nephrotic syndrome in children, in addition to infection and venous thromboembolism. Risk factors for AKI include steroid-resistant nephrotic syndrome, infection, and nephrotoxic medication exposure. Children with AKI have longer hospital lengths of stay and increased need for intensive care unit admission (doi: https://doi.org/10.2215/CJN.06620615). Some biochemical tests have been carried out, but in the introduction has no justification. A short paragraph will contribute to the manuscript.

Point 2: Immunology Analyses (doi: 10.1080/20469047.2017.1374003).

Point 3: Blood coagulation tests (doi: 10.1159/000168196). MATERIAL AND METHODS

Point 4: The way biochemical analyses were performed is not indicated in the material and methods.

Point 5: How was the biopsy performed? How was histology done?

Point 6: Insert the statistical analysis method used. The use of statistical symbols in table 1 can facilitate the identification of non-normal parameters.

Point 7: (Page 2. Line 52): Attach the consent form as complementary material.

Point 8: Include participant characteristics in the materials and methods. Example (doi:10.1001/jamanetworkopen.2022.28701): “NEPTUNE included 166 participants (32 children, 29 adolescents, and 105 adults); FSGS-CT included 132 (42 children, 48 adolescents, and 42 adults); KRN included 184 (53 children, 25 adolescents, and 106 adults). Overall, the study included 127 (26%) children, 102 (21%) adolescents, and 253 (52%) adults, including 215 (45%) female participants and 138 (29%) who identified as Black, 98 (20%) who identified as Hispanic, and 275 (57%) who identified as White.” RESULTS

Point 9: Examine these findings in Table 1. The authors will need support in the literature on why an only patient with focal segmental glomerulosclerosis showed a lower proteinuria level than patients V-4 and V-7. Or, alternatively, why patients V-4 and V-7 do not have focal segmental glomerulosclerosis with high levels of proteinuria? “-Patient V-4, age 13, with focal segmental glomerulosclerosis and proteinuria of 300 mg/dl. -Patient V-7, aged 11, without focal segmental glomerulosclerosis and proteinuria of 700 mg/dl. -Patient V-8 R, aged 8, without focal segmental glomerulosclerosis and proteinuria of 650 mg/dl.” Importance: Focal segmental glomerulosclerosis (FSGS) is one of the most common causes of end-stage kidney disease (ESKD), accounting for 10% to 15% of pediatric and 3% of adult ESKD population in the United States (doi:10.1001/jamanetworkopen.2022.28701). In addition: Proteinuria is a hallmark of FSGS as well as other primary and secondary glomerular disorders (doi: 10.3389/fmed.2018.00098).

Point 10: Figure 2: Image resolution improvement; images can be enlarged and brightness better.

Point 11: Suggestion: additional biochemical analyses to be added to Table 1. - To detect and identify proteinuria, use the urine albumin/creatinine ratio (ACR). Albuminto-creatinine ratio (ACR) is the first method of preference to detect elevated protein (https://www.kidney.org/content/kidney-failure-risk-factor-urine-albumin-to-creatinineration-uacr) - The urinary protein/creatinine ratio has been used in Nephrology practice for the followup of glomerular diseases (https://doi.org/10.1186/s12882-019-1486-8) REFERENCES Point

12: I did not see reference 16 in the text.

Point 13: I did not see reference 18 in the text.

Point 14: Standardize whether or not to insert the "DOI" into the references.

Author Response

Reviewer 2

The manuscript presented by the Authors entitled "Exome Sequencing Revealed a Novel Splice Site Variant in the 2 CRB2 Gene Underlying Nephrotic Syndrome" has potential but can be improved. The objectives are stated clearly. The results presented are relevant. The bibliography is up-to-date. The following are my comments describing these issues.

INTRODUCTION

The Authors should deepen these aspects in the introduction:

Point 1: Various complications have been reported in children with NS, including infections, thromboembolism, hypovolemia, and acute kidney injury (doi: 10.2215/CJN.06620615). AKI is common in children hospitalized with nephrotic syndrome and should be deemed the third major complication of nephrotic syndrome in children, in addition to infection and venous thromboembolism. Risk factors for AKI include steroid-resistant nephrotic syndrome, infection, and nephrotoxic medication exposure. Children with AKI have longer hospital lengths of stay and increased need for intensive care unit admission (doi: https://doi.org/10.2215/CJN.06620615). Some biochemical tests have been carried out, but in the introduction has no justification. A short paragraph will contribute to the manuscript.

Response: Thank you for the valuable comments. We have added a paragraph related to complications associated with nephrotic syndrome.

Point 2: Immunology Analyses (doi: 10.1080/20469047.2017.1374003).

Response: Added

Point 3: Blood coagulation tests (doi: 10.1159/000168196). MATERIAL AND METHODS

Response: Added

Point 4: The way biochemical analyses were performed is not indicated in the material and methods.

Response: Added

Point 5: How was the biopsy performed? How was histology done?

Response: The following lines have been added in materials and methods section. Biopsy was performed under general anesthesia under strict aseptic measures in operation theatre. Automated Trucut biopsy needle was used to take biopsy under ultrasound guidance. Two samples were taken, one for immunofluorescence and one for histopathology.

Point 6: Insert the statistical analysis method used. The use of statistical symbols in table 1 can facilitate the identification of non-normal parameters.

Response: We have only three siblings in a single family; therefore, we have not performed statistical analysis. We have only used reference values to test whether the values of biochemical tests of the patients are in normal range.

Point 7: (Page 2. Line 52): Attach the consent form as complementary material.

Response: Thank you for asking for the consent form. We have attached the consent forms provided by the affected family.

Point 8: Include participant characteristics in the materials and methods. Example (doi:10.1001/jamanetworkopen.2022.28701): “NEPTUNE included 166 participants (32 children, 29 adolescents, and 105 adults); FSGS-CT included 132 (42 children, 48 adolescents, and 42 adults); KRN included 184 (53 children, 25 adolescents, and 106 adults). Overall, the study included 127 (26%) children, 102 (21%) adolescents, and 253 (52%) adults, including 215 (45%) female participants and 138 (29%) who identified as Black, 98 (20%) who identified as Hispanic, and 275 (57%) who identified as White.” RESULTS

Point 9: Examine these findings in Table 1. The authors will need support in the literature on why an only patient with focal segmental glomerulosclerosis showed a lower proteinuria level than patients V-4 and V-7. Or, alternatively, why patients V-4 and V-7 do not have focal segmental glomerulosclerosis with high levels of proteinuria? “-Patient V-4, age 13, with focal segmental glomerulosclerosis and proteinuria of 300 mg/dl. -Patient V-7, aged 11, without focal segmental glomerulosclerosis and proteinuria of 700 mg/dl. -Patient V-8 R, aged 8, without focal segmental glomerulosclerosis and proteinuria of 650 mg/dl.” Importance: Focal segmental glomerulosclerosis (FSGS) is one of the most common causes of end-stage kidney disease (ESKD), accounting for 10% to 15% of pediatric and 3% of adult ESKD population in the United States (doi:10.1001/jamanetworkopen.2022.28701). In addition: Proteinuria is a hallmark of FSGS as well as other primary and secondary glomerular disorders (doi: 10.3389/fmed.2018.00098).

Response: As these are single time reports. Proteinuria in these patients are checked serially, ideally daily. She might be in partial remission at the time of testing.

Point 10: Figure 2: Image resolution improvement; images can be enlarged and brightness better.

Response: A revised figure with better quality has been added in the revised manuscript.

Point 11: Suggestion: additional biochemical analyses to be added to Table 1. - To detect and identify proteinuria, use the urine albumin/creatinine ratio (ACR). Albuminto-creatinine ratio (ACR) is the first method of preference to detect elevated protein (https://www.kidney.org/content/kidney-failure-risk-factor-urine-albumin-to-creatinineration-uacr) - The urinary protein/creatinine ratio has been used in Nephrology practice for the followup of glomerular diseases (https://doi.org/10.1186/s12882-019-1486-8)

Response: We tried to follow up the patients but unfortunately, they declined to provide samples for further tests as recently their father and one of the siblings died while traveling for routine checkups.

REFERENCES Point

12: I did not see reference 16 in the text.

Response: Added

Point 13: I did not see reference 18 in the text.

Response: Added

Point 14: Standardize whether or not to insert the "DOI" into the references.

Response: Corrected